# Fluctuation of Plasmonically Induced Transparency Peaks within Multi-Rectangle Resonators

**DOI:** 10.3390/s23010226

**Published:** 2022-12-26

**Authors:** Ruoyu Pei, Dongdong Liu, Qun Zhang, Zhe Shi, Yan Sun, Xi Liu, Jicheng Wang

**Affiliations:** 1Xinjiang Laboratory of Phase Transitions and Microstructures in Condensed Matters, College of Physical Science and Technology, Yili Normal University, Yining 835000, China; 2State Key Laboratory of Millimeter Waves, Southeast University, Nanjing 210096, China; 3School of Physics and New Energy, Xuzhou University of Technology, Xuzhou 221018, China; 4School of Science, Jiangnan University, Wuxi 214122, China

**Keywords:** plasmon-induced transparency, metal–insulator–metal, one falls another rises, nanoscale optical switches, sensors

## Abstract

Numerical investigations were conducted of the plasmonically induced transparency (PIT) effect observed in a metal–insulator–metal waveguide coupled to asymmetric three-rectangle resonators, wherein, of the two PIT peaks that were generated, one PIT peak fell while the other PIT peak rose. PIT has been widely studied due to its sensing, slow light, and nonlinear effects, and it has a high potential for use in optical communication systems. To gain a better understanding of the PIT effect in multi-rectangle resonators, its corresponding properties, effects, and performance were numerically investigated based on PIT peak fluctuations. By modifying geometric parameters and filling dielectrics, we not only realized the off-to-on PIT optical response within single or double peaks but also obtained the peak fluctuation. Furthermore, our findings were found to be consistent with those of finite element simulations. These proposed structures have wide potential for use in sensing applications.

## 1. Introduction

Surface plasmon polaritons (SPPs) are transverse electromagnetic waves [1] that result from collective oscillations between photons and free electrons, and they can be confined and spread along metal and dielectric surfaces [2,3,4]. SPPs have recently received increased attention for two reasons. First, SPPs can overcome light diffraction [5]. Second, they can manipulate light at sub-wavelength scales [6,7]. To date, numerous numerical simulations and experiments have been conducted on optical elements based on SPPs. From these studies, two representative plasmonic structures, insulator–metal–insulator (IMI) structures and metal–insulator–metal (MIM) structures, were found to be essential as optical components. Furthermore, SPPs-based IMI structures positively contribute to energy loss over long transmission distances. Because the light is limited to sub-wavelength scales, IMI structures are not suitable as high-integration-degree optical components. However, the unique structure of MIMs allows for deep sub-wavelength scales and high group velocity over a wide frequency range, in addition to also providing high optical confinement and acceptable propagation length [8]. As a result, MIM structures can be used in a variety of applications, including in optical filters [9], optical switches [10,11,12,13], Mach–Zehnder interferometers [14,15], demultiplexers [16], and nanosensors [17,18,19,20,21].

EIT is a well-known physical phenomenon that occurs in a coherently driven atomic system and can significantly suppress the absorption behavior of detection light within a narrow spectral range, resulting in a narrow, transparent window in the transmission spectrum [22,23]. Therefore, EIT is a promising candidate for a variety of applications such as signal processing [24,25], nanobiosensors [26], and optical data storage [27]. Plasmonically induced transparency (PIT) is a novel phenomenon with some properties that allow it to outperform EIT in some ways. Furthermore, because of its room-temperature manipulation, large bandwidth, and integration with nanoplasmonic circuits [28,29], PIT offers convenient and selective operation options. For example, EIT has been demonstrated through interaction with a wide silver line as a bright pattern and a pair of narrow silver lines as a dark pattern. The PIT phenomenon was caused by near-field coupling between light and dark modes in this structure. Liu et al. [30] developed an MIM with multiple rectangular resonators. Based on this, multiple plasma-induced transparent peaks and a controllable number of transparent peaks were achieved by adjusting the geometric parameters [31,32,33,34,35,36,37,38,39].

Previously, only a few studies have looked into the simultaneous and opposed adjustment of PIT peaks in same-order modes. As a result, modified modes based on traditional symmetric resonators were proposed [40]. In contrast to the plasma-induced transparency of the previous traditional structure, asymmetrical multi-rectangle resonators can not only achieve single or multiple induced transparency peaks but also realize energy-coupled path change [41,42,43,44,45,46]. As a result, the modulation parameters cause two transmission peaks in the same state to alternate.

In this study, a classic PIT structure was used as the basis for forming a MIM bus waveguide coupled to asymmetric three-rectangle resonators, and the corresponding influences were numerically investigated. The relationships between the PIT optical response and the widths, heights, refractive index, and location of the double transmission peaks of the three cavities were also thoroughly investigated. According to our findings, peak fluctuation in one same-order mode was observed, which was caused by the change in the energy coupling pathway. Numerical simulations performed by the commercial finite element method (FEM) software COMSOL Multiphysics demonstrated our theoretical analysis.

## 2. Model and Theoretical Analysis

Shown in Figure 1 is the schematic representation of the two-dimensional plasmonic filter structure comprising an MIM waveguide with side-coupled multi-rectangle resonators. The width of the MIM waveguide is set to *w* = 50 nm. The input and output ports are numbered 1 and 2, respectively. The insulators in the white areas are assumed to be air, and its refractive index is 1 (*ε_i_* = 1). Cavity I, cavity II, and cavity III are the names of the three resonators. The rectangular resonator’s height, width, and coupling distance are *h_i_* (*i* = 1, 2, 3), *w_i_* (*i* = 1, 2, 3), and *t*, respectively. The distances from the three cavities to the main waveguide are all set as *t* = 20 nm. The distance between cavity I and cavity II is set as *d* = 20 nm. As shown in Figure 1, silver is used as the metal to fill the dark area. Silver’s relative permittivity is proportional to the frequency of incident light, and its dielectric constant can be calculated using the well-known Drude model [47]:(1)$$\varepsilon_{m}\left(\omega \right)=\varepsilon_{\infty }-\frac{\omega_{p}^{2}}{\omega \left(\omega +i\gamma \right)}$$

In Equation (1), *ε_∞_* = 3.7 denotes the infinite-frequency dielectric constant, *γ* = 2.73 × 10^13^ Hz denotes the electron collision frequency, *ω_p_* = 1.38 × 10^16^ Hz denotes the bulk plasma frequency, and the angular frequency of incident electromagnetic radiation is denoted by *ω*. All structural slits mentioned in this paper have a width of 50 nm. Because the incident wavelength is greater than *w*, only the basic plasmon mode TM_0_ can exist in the waveguide structure. The constant for incident light propagation can be calculated using the following equation [48]:(2)$$\tanh\left(\frac{w\sqrt{\beta^{2}-k_{0}^{2}\varepsilon_{i}}}{2}\right)=\frac{-\varepsilon_{i}\sqrt{\beta^{2}-k_{0}^{2}\varepsilon_{m}\left(\omega \right)}}{\varepsilon_{m}\left(\omega \right)\sqrt{\beta^{2}-k_{0}^{2}\varepsilon_{i}}}$$

In Equation (2), *ε_m_* and *ε_i_* are the dielectric constants of silver and air, respectively. Under a vacuum, the wave vector of light is *λ*_0_. It is possible to obtain *n_eff_* = *β*/*k*_0_ using Equation (2). Figure 2 depicts the real part of *n_eff_* as a function of *d* and *λ*. Once *w* is determined, Re (*n_eff_*) can be kept constant for various incident wavelengths.

When incident light passes through this designed structure, one portion is coupled into the resonator while the remainder is reflected and transmitted. Coupled-mode theory can be used to describe the transmission spectrum near the resonant mode, and transmission *T* can be mathematically expressed as in Equation (3):(3)$$T(\omega )=\frac{{\left(\omega -\omega_{0}\right)}^{2}+{\left(1/\tau_{i}\right)}^{2}}{{\left(\omega -\omega_{0}\right)}^{2}+{\left(1/\tau_{i}+1/\tau_{\omega }\right)}^{2}}$$

In Equation (3), *ω* and *ω*_0_ represent the incident light frequency and the resonance frequency, respectively. The decay rates of the nanocavity internal loss and power that escapes through the waveguide are represented by 1/*τ_i_* and 1/*τ_ω_*, respectively. *T_min_* = (1/*τ_i_*)^2^/(1/*τ_i_* + 1/*τ_ω_*)^2^ at the resonance frequency *ω*_0_ can thus be used to calculate the minimum transmission ratio.

## 3. Results and Discussions

As shown in Figure 3a, two valleys were seen at *λ*_0_ = 1301 and 880 nm, representing the first- and second-order modes, respectively, with transmission spectra ranging from 750 nm to 1600 nm. The magnetic field intensity distributions at *λ*_0_ = 880 and *λ*_0_ = 1301 nm are shown in Figure 3b,c, respectively. The majority of the energy in the incident light at *λ*_0_ = 880 nm was coupled into cavity III and reflected back and forth, as shown in Figure 3b. When the wavelength of the incident light was increased to 1301 nm, the majority of the energy was coupled into and reflected back and forth between the three rectangular cavities. These results indicate that a small amount of energy can be transmitted in this mode, which is advantageous for band-stop filter applications.

To investigate the phenomenon of PIT tunability, the effect of changing the cavity III parameters on the transmission spectrum was considered, while the parameters of cavities I and II remained constant. Thus, the width, height, and refractive index of cavity III were gradually altered while the other parameters were set as described above.

The transmission spectrum as a function of cavity III width *w*_3_ was considered in the first stage. The PIT window was generated by the destructive interference between cavity III and the other two cavities when *w*_3_ ≠ *w*_1_ + *w*_2_ + *d* and *w*_3_ was varied from 1020, 1040, 1060, and 1080 to 1100 nm, as shown in Figure 4a. As the width *w*_3_ increased, the transmission dip in the first-order mode split into two dips, indicating the appearance of the PIT window. With the increase in destructive interference between cavity III and the other two cavities, the PIT window grew wider and increasingly visible. As the *w*_3_ was increased, the PIT transmission peaks exhibited a redshift effect, and the transparency windows showed a progressively off-to-on optical response.

Second, transmission spectra were obtained by varying the height *h*_3_ of cavity III, as illustrated in Figure 4b. When the height parameter *h*_3_ was increased in 10 nm increments from 90 to 130 nm, an obvious off-to-on multimode PIT response and transmission spectrum blue shift were seen. The *Q* factor of the PIT windows was calculated as λ_0_/Δλ, where λ_0_ represents the transparency peak wavelength of the transparency window and represents the full width at half maximum (FWHM) of the transparency window. When *h*_3_ was increased, the *Q* value decreased significantly, indicating that the height *h*_3_ of cavity III had a significant impact on the *Q* value of the fundamental mode PIT window. This significant feature could be used to resize the *Q* of the PIT resonance. Various refractive indexes of the filling medium inside cavity III, ranging from 1.0 to 1.08 with a step size of 0.02, were tested to gain an understanding of the effect of changing the refractive index. A series of transmission spectra for adjusting the refractive index are shown in Figure 4c. The transmission spectrum and variation pattern were similar to those shown in Figure 4a,c. Both graphs showed increased transmission and redshifting as the corresponding parameters were increased. As a result, increasing the value of the refractive index *n*_3_ was essentially equivalent to increasing the width *w*_3_.

The first-order model transparent window is depicted in Figure 5 as *h*_3_ = 120 nm, which corresponds to the magnetic field intensity distribution |*Hz|*^2^.

As shown in Figure 5b, when the wavelength of incident light was 1251 nm, the majority of the energy was stored in the rectangular cavity III, causing a dip in the transmission spectrum. While *λ* = 1309 nm, the majority of the energy was coupled to cavities I and II, with only small amounts of energy entering cavity III, as shown in Figure 5d. Little energy was distributed into three rectangular cavities when *λ* = 1281 nm, as shown in Figure 5c. Meanwhile, SPPs could pass through the waveguide to the output port and acted as a PIT “on” state.

The structure could be easily expanded into a double PIT system by adjusting the parameters of two cavities at the same time. Figure 6 depicts a series of transmission spectra after synchronous adjustment of cavities I and II.

The height *h*_1_ of cavity I and the width *w*_2_ of cavity II were varied from 100 to 140 nm and 520 to 600 nm with a step of 10 nm and 20 nm, respectively. Other structural parameters remained constant. In the first-order mode, two new transmission peaks appeared, which can be attributed to the double PIT light response process. The light response of the transparent window gradually turned on as *h*_1_ was increased from 100 to 140 nm and *w*_1_ increased from 520 to 600 nm. In the first-order mode, both the left and right peaks exhibited redshift behavior.

Figure 7 depicts the three dips and two peaks in the first-order mode, which correspond to the magnetic field intensity distribution |*Hz|*^2^. Because the energy was almost confined to cavity III, the two field distribution figures are similar to those in Figure 5b and Figure 7d. As illustrated in Figure 7b (*λ* = 1270 nm) and Figure 7f (*λ* = 1549 nm), the majority of the energy was coupled into cavities I and II, implying that the right and left new dips in the first-order mode could be controlled by resizing the *h*_1_ of cavity I and the *w*_2_ of cavity II, respectively. The SPPs bounced between the cavities and the bus waveguide. The system is, therefore, a classic Fabry–Pérot resonator, and the new dip was probably caused by adjusting the height of cavity I and the width of cavity II. As Figure 7e (*λ* = 1507 nm) shows, part of the energy could be coupled into cavities II and III, and SPPs were able to travel through the bus waveguide to port 2, indicating the “on” state of the PIT.

After resizing the structure of cavity II and simultaneously adjusting the parameters of cavities I and III, we observed two PIT peaks in the first-order mode, wherein one peak was rising while the other was falling.

The values for the height *h*_2_ and width *w*_2_ of cavity II were swapped, but the other parameters remained constant. The height *h*_2_ was set to 500 nm, and the width *w*_2_ to 90 nm. Figure 8a depicts the structure. The height parameters of cavities I and III, *h*_1_ and *h*_3_, were then varied synchronously from 50 to 90 nm in 10 nm steps while other variables remained constant. Figure 8b depicts the transmission spectrum and variation rule. The system induced two transparent peaks between 1200 and 1600 nm, which exhibited the phenomenon of one rising and the other falling. As the height of cavities I and III was increased in first-order mode, the right peak decreased while the left peak rose. The height of the two transmission peaks was similar when *h*_1_ = *h*_2_ = 70 nm (Figure 8b). When *h*_1_ = *h*_2_ = 90 nm, the left peak in this region disappeared.

When the magnetic field distributions of the right dip in Figure 9a,b (*h*_1_ = *h*_2_ = 60 nm) (*h*_1_ = *h*_2_ = 80 nm) were compared, the energy transitioned from being almost uncoupled to cavity I to becoming progressively coupled to cavity I as *h*_1_ and *h*_2_ increased. It was possible to demonstrate that changing two parameters at the same time altered the energy coupling paths. As a result, the two transmission peaks of the PIT structure could be adjusted in opposing directions.

As shown in Figure 10, the shifting behaviors allowed cavity II to partially overlap with cavity I. Other conditions remained constant, and the effect of changing different cavity II parameters on the transmission spectrum was investigated. In this case, changing the height or width of cavities I and II had no effect on the transmission spectrum. The energy was primarily confined to cavity III because it was much wider than cavity I.

Based on the information presented above, the width of cavity III was reduced to *w*_1_ = *w*_3_ = 500 nm to investigate the effect of cavity II on the transmission spectrum. Figure 11a depicts the structure and parameters.

The height of cavity II was adjusted correspondingly to the height adjustment of bulge *b* from Figure 11a. The height of cavity II parameter *b* was set to 390, 400, 410, 420, and 430 nm. Other variables were held constant. In the two transmission peaks of the second-order mode, the phenomenon of one rising and the other falling was also achieved. As shown in Figure 11b, the left transmission peak in the second-order mode gradually decreased, whereas the right transmission peak in the second-order mode became more visible as *b* was increased. During the process of *b* increasing, the two transmission peaks also exhibited a slight redshift.

By comparing the magnetic field distributions of the peaks in Figure 12a,b, we can see that the energy gradually shifted from being primarily coupled into cavities II and III to being coupled into cavities I and III. The energy coupling path in the structure could, therefore, be changed by adjusting only one parameter of cavity II. As a result, the two transparent peaks in the same mode responded differently. These results provide a theoretical basis for designing optical biosensors or other highly integrated optical devices.

## 4. Conclusions

In this study, the transmission characteristics and rules of SPPs in different structures composed of MIM waveguides and three-rectangle resonators were systematically investigated. Finite element numerical simulation and theoretical analysis were used to test the hypothesis and determine the rule of different modes. Changing the geometrical parameters of the cavities, such as the width and height of a rectangular cavity or the refractive index, caused an off-to-on multi-wavelength PIT optical response and made it possible to control the response intensity. On this basis, a novel phenomenon for two transmission peaks of the same-order mode wherein one peak rises and one falls was discovered and studied. This phenomenon, caused by a change in the energy coupling path, can be realized by changing one or two parameters in the various structures discussed in the paper. As a result, two transmission peaks in the same mode can be simultaneously adjusted in opposite directions at the same time via the structural parameters, allowing for more flexible filtering of different bands. The various structures and rules discussed in this paper can be applied to actively tunable sensors, paving the way for actively tunable sensor experiments.

## Figures and Tables

**Figure 1 sensors-23-00226-f001:**
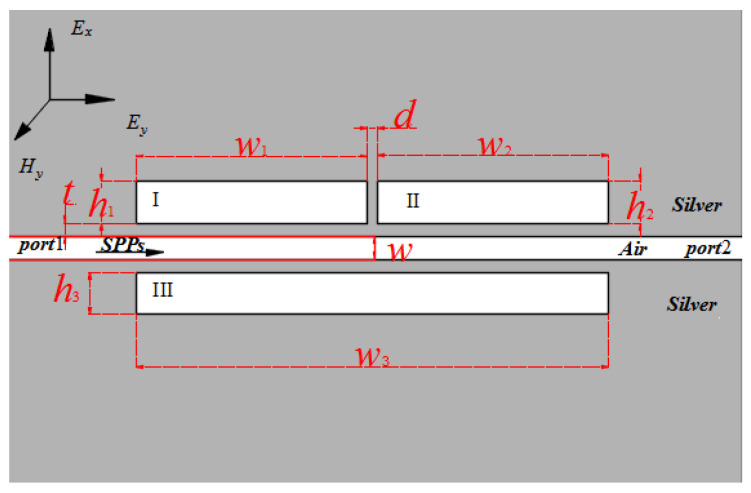
Structure diagram of a MIM structure with three rectangular resonators.

**Figure 2 sensors-23-00226-f002:**
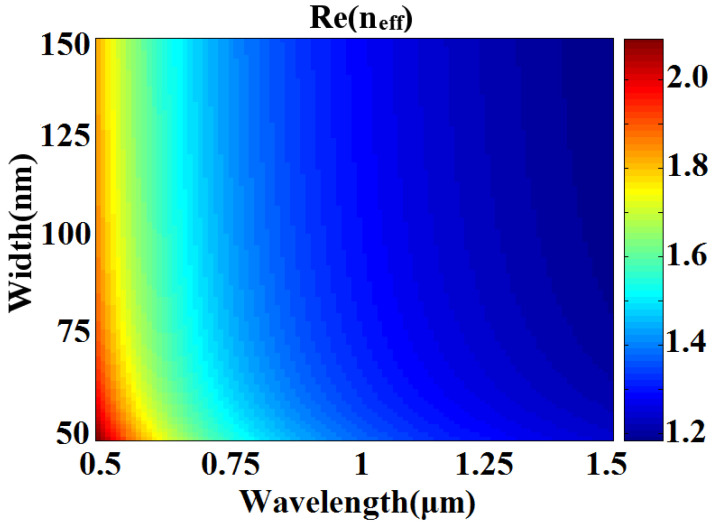
Real part of the effective refractive index *n_eff_* versus the incident wavelength *λ* and the width *w* in the MIM waveguide.

**Figure 3 sensors-23-00226-f003:**
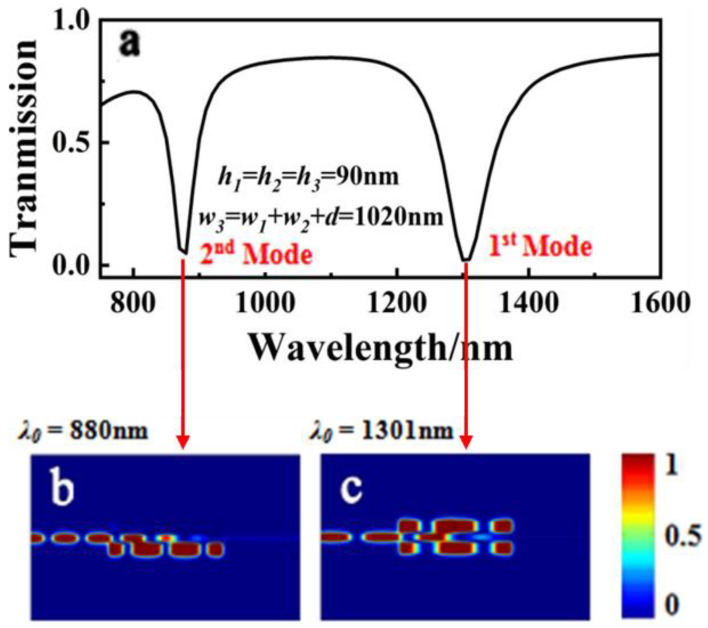
(**a**) Transmission spectrum of the MIM waveguide with the three-rectangle resonators shown in Figure 1 when *h*_1_ = *h*_2_ = *h*_3_ = 90 nm and *w*_3_ = *w*_1_ + *w*_2_ + *d* = 1020 nm. The field distribution |*H_z_*|^2^ of the structure at an incident wavelength of (**b**) 880 nm and (**c**) 1301 nm.

**Figure 4 sensors-23-00226-f004:**
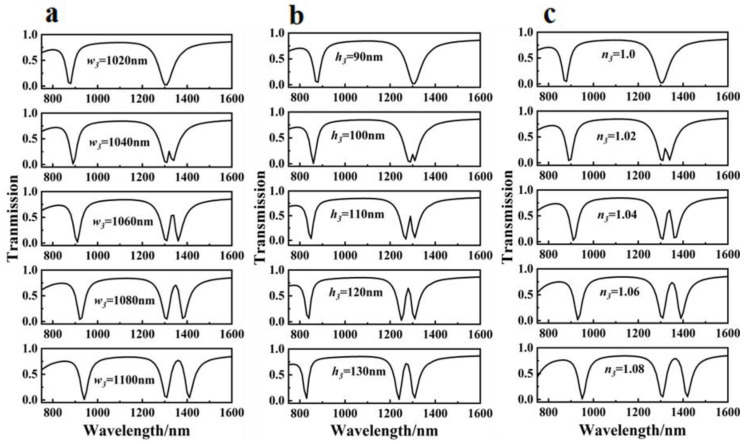
Off-to-on transmission spectra of three-rectangle-resonator PIT system with different (**a**) *w*_3_, (**b**) *h*_3_, and (**c**) *n*_3_.

**Figure 5 sensors-23-00226-f005:**
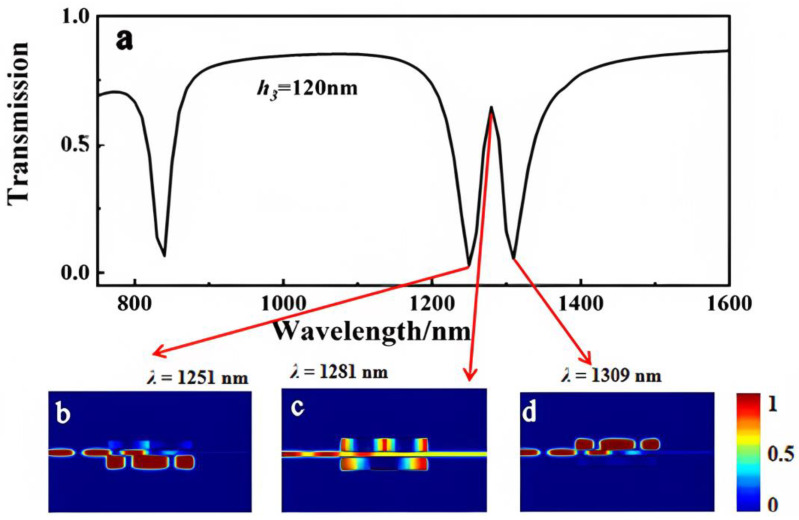
(**a**) Transmission spectrum of the MIM waveguide with the three-rectangle resonators shown in Figure 1 when *h*_3_ = 120 nm. The field distribution |*Hz*|^2^ of the structure at the incident wavelengths of (**b**) 1251 nm, (**c**) 1281 nm, and (**d**) 1309 nm.

**Figure 6 sensors-23-00226-f006:**
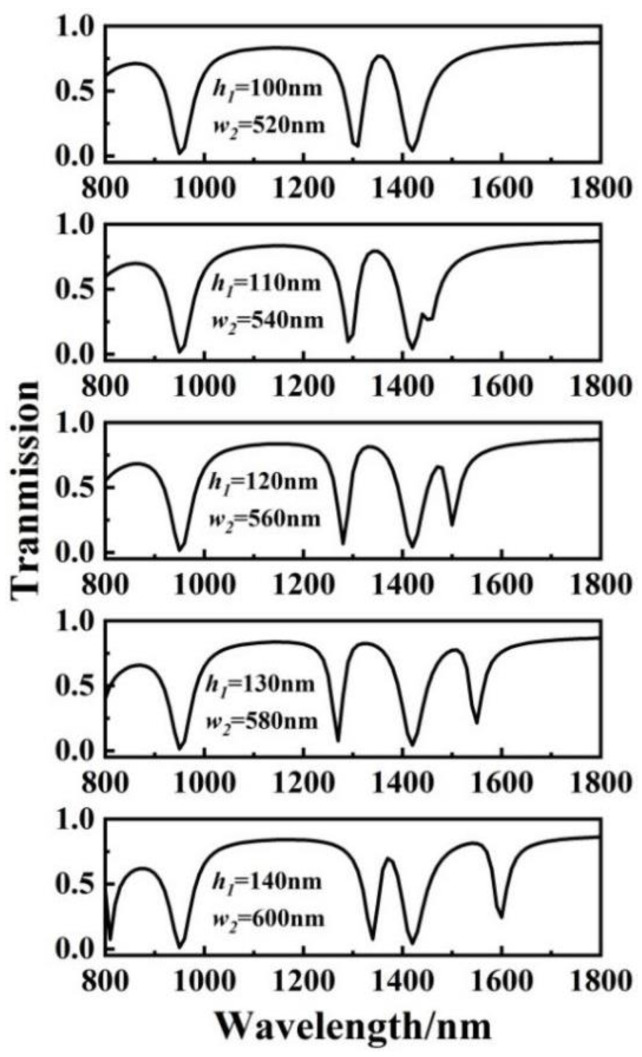
Off-to-on transmission spectra of the three-rectangle-resonator PIT system with different values of *h*_1_ and *w*_2_.

**Figure 7 sensors-23-00226-f007:**
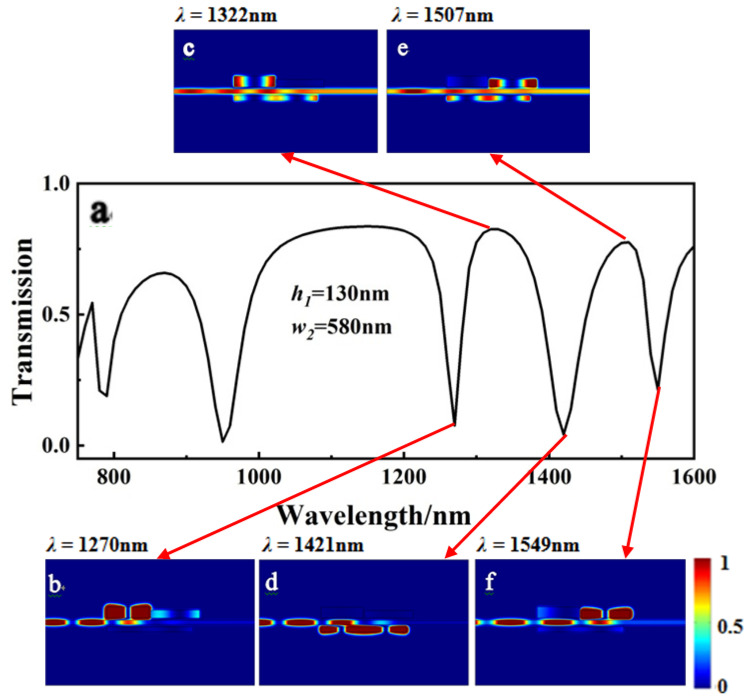
(**a**) Transmission spectrum of the MIM waveguide with the three-rectangle resonators shown in Figure 1 when *h*_1_ and *w*_2_ were synchronously changed. The field distribution *|Hz|*^2^ of the structure at incident wavelengths of (**b**) 1270 nm, (**c**) 1322 nm, (**d**) 1421 nm, (**e**) 1507 nm, and (**f**) 1549 nm.

**Figure 8 sensors-23-00226-f008:**
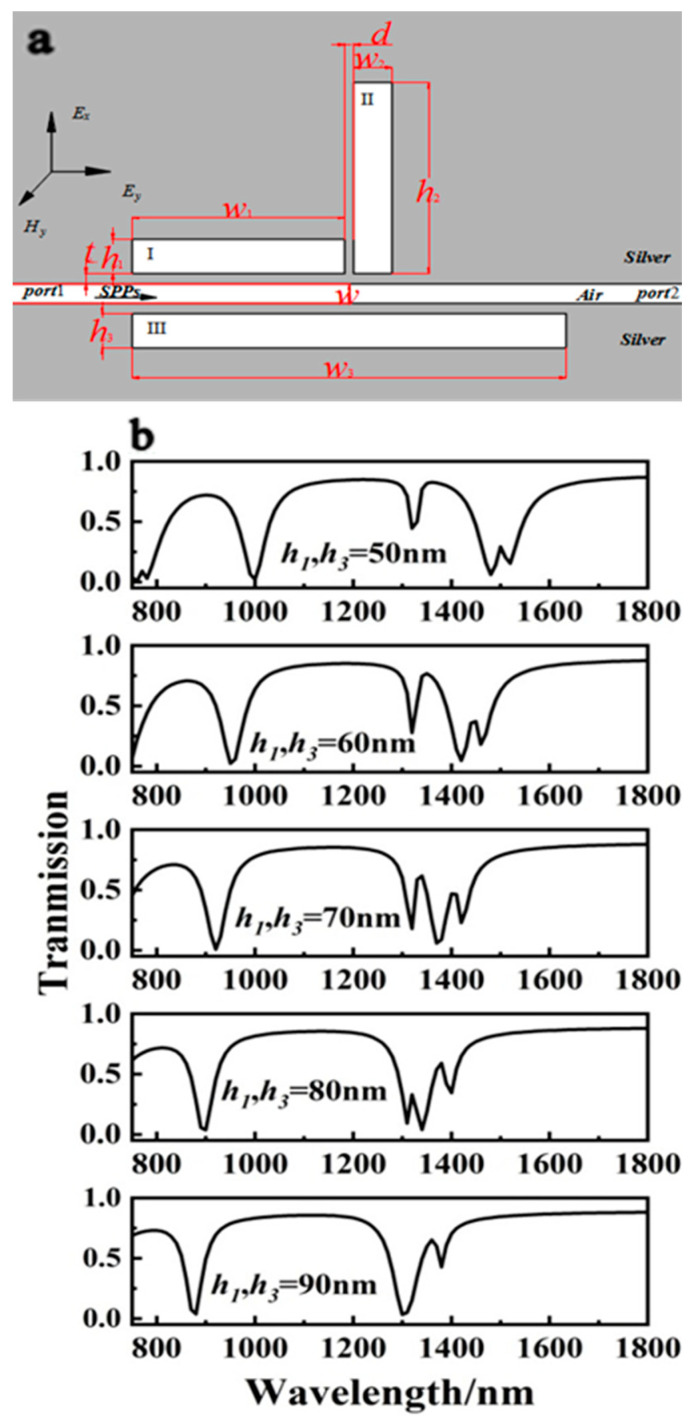
(**a**) Structure diagram of MIM with three rectangular resonators after adjusting cavity II. (**b**) Off-to-on transmission spectra of the four-rectangle-resonator PIT system with different *h*_1_ and *h*_3_ values.

**Figure 9 sensors-23-00226-f009:**
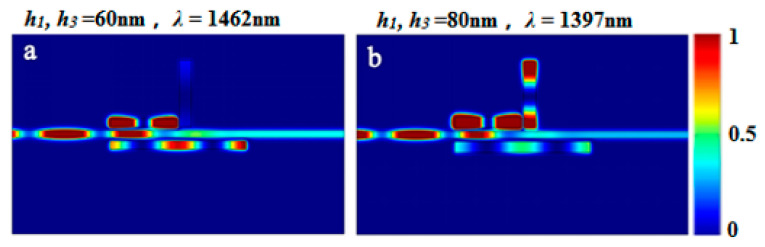
The field distribution *|Hz|*^2^ of the structure at (**a**) incident wavelength 1462 nm when *h*_1_, *h*_3_ = 60 nm; (**b**) incident wavelength 1397 nm when h1, h3 = 80 nm.

**Figure 10 sensors-23-00226-f010:**
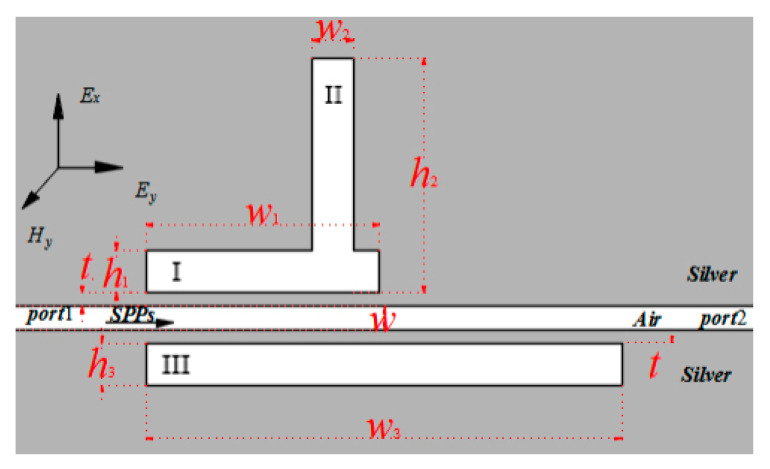
Structure diagram of MIM with three rectangular resonators after cavities I and II were partially combined.

**Figure 11 sensors-23-00226-f011:**
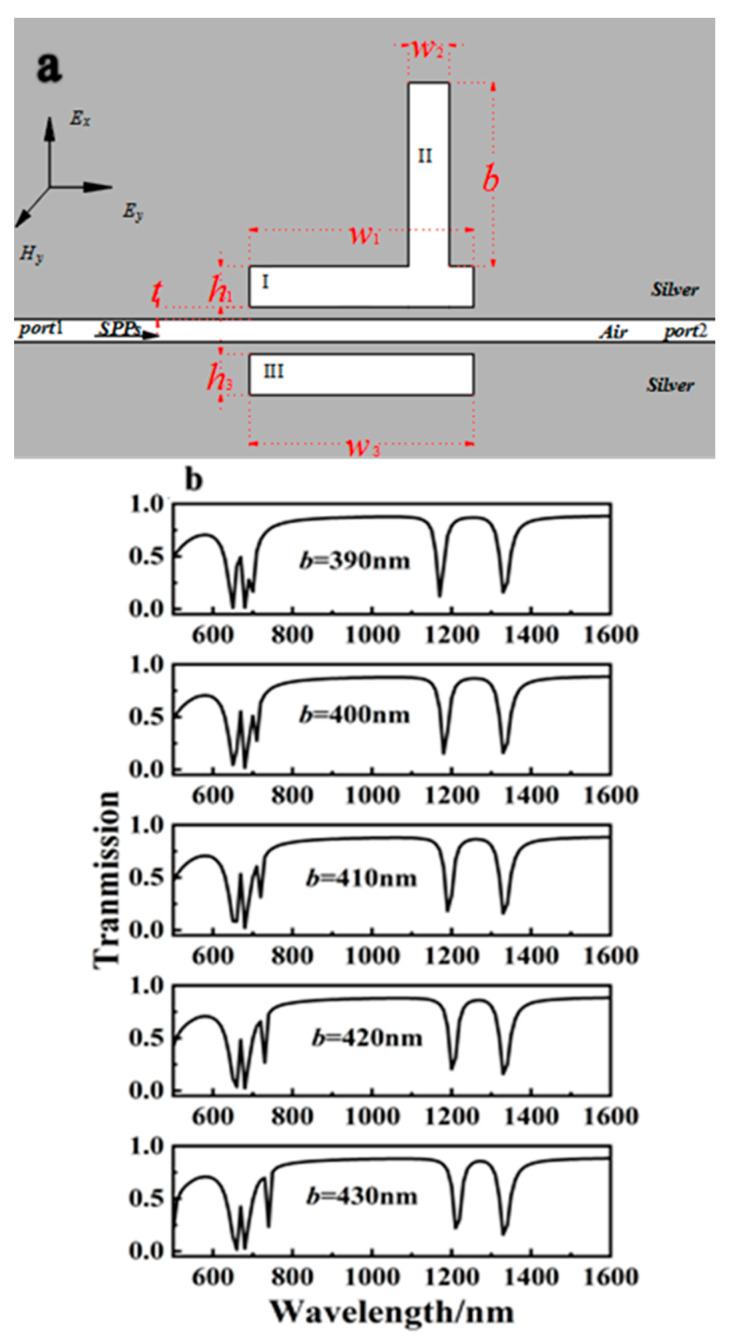
(**a**) Structure diagram of a MIM with three rectangular resonators after cavities I and II were partially combined and *w*_1_ = *w*_3_ = 500 nm. (**b**) Off-to-on transmission spectra of the three-rectangle-resonator PIT system.

**Figure 12 sensors-23-00226-f012:**
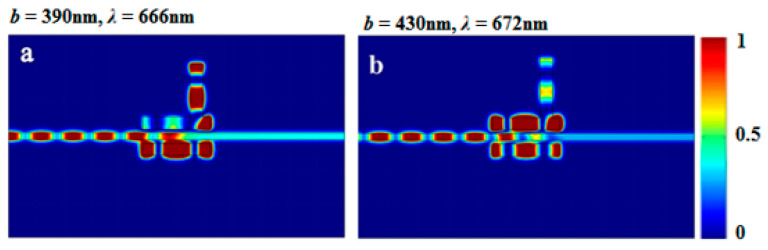
(**a**) The field distribution |*H*_z_|^2^ of the structure at (**a**) incident wavelength 666 nm when *b* = 390 nm; (**b**) incident wavelength 672 nm when *b* = 430 nm.

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
