# Peer review of "Fluctuation of Plasmonically Induced Transparency Peaks within Multi-Rectangle Resonators"

_sensors, 2022, doi:10.3390/s23010226_

Round 1

Reviewer 1 Report

This work reported numerical investigation of the plasmonically induced transparency PIT effect in multi-rectangle resonators. Additionally, to get a better insight of the PIT effect, corresponding properties, effects, and performance are investigated as well. Noticeably, in sharp contrast to previous reported investigations, asymmetry multi-rectangle resonators are detailed investigated through the proposed numerical model by author. The outcomes of this manuscript are potentially interesting, whereas there are still some concerns related to the simulation details, which should be addressed by the authors. In general, this research work is well organized and presented, I recommend the manuscript to be published on SENSORS after minor revisions. The detail comments are listed below:

1. The highlights parts should be as concise as possible.

2. Page 5: “The first-order model transparent window is depicted in Fig. 5 as h3 = 120 nm, which corresponds to the magnetic field intensity distribution |Hz|2. As shown in Fig. 5(b), when the incident light is at 1251 nm, the majority of the energy is stored in the rectangular cavity III, causing a dip in the transmission spectrum. While λ = 1309 nm, the majority of the energy is coupled to cavities I and II, with only small amounts of energy entering cavity III, as shown in Fig. 5(d). Energy distrusts in all three rectangular cavities when λ = 1281 nm, as shown in Fig. 5(c). Meanwhile, SPP could access the output port via the waveguide.” Why? What is the physical mechanism here?

3. Page 5: “The height of cavity I h1 is set to 100, 110, 120, 130, and 140 nm, and the width of cavity II w2 is set to 520 to 600 nm with a 20 nm step.” can be re-wrote as “The height of cavity I h1 and the width of cavity II w2 are varied from 100 to 140 nm, 520 to 600 nm with a step of 10 nm and 20 nm, respectively.”

4. Page 7: “As shown in Fig. 8, the shifting behaviors allow cavity II to partially overlap with cavity I.” Which should be Fig. 10. The cavity I, II and III should be signed in this figure as well.

5. Page 7: “During the process of b increasing, the two transmission peaks also exhibit a slight red shift.” Why? What is the physical mechanism here?

6. Page 9: “The phenomenon caused by a change in the energy coupling path can be realized by changing one or two parameters in the various structures discussed in the paper.” To be revised.

Author Response

  1. The highlights parts should be as concise as possible.

Thanks for your suggestion. We have condensed the highlights of the article on page 2. Some repeated descriptions during the experiment were removed.

Reply: We deleted “Within EIT, not only can M-class double pulses coherent control be obtained, but also sharp resonance and steep dispersion.”, “PIT offers convenient and selective operation options.” and “As a result, modified modes based on traditional symmetric resonators were proposed.” in the highlights of the article.

  1. Page 5: “The first-order model transparent window is depicted in Fig. 5 as h3 = 120 nm, which corresponds to the magnetic field intensity distribution |Hz|2. As shown in Fig. 5(b), when the incident light is at 1251 nm, the majority of the energy is stored in the rectangular cavity III, causing a dip in the transmission spectrum. While λ = 1309 nm, the majority of the energy is coupled to cavities I and II, with only small amounts of energy entering cavity III, as shown in Fig. 5(d). Energy distrusts in all three rectangular cavities when λ = 1281 nm, as shown in Fig. 5(c). Meanwhile, SPP could access the output port via the waveguide.” Why? What is the physical mechanism here?

Reply: When the incident light λ is at 1251nm, it would be coupled into the cavity III and be reflected back and forth. Little energy would be transmitted, so the model can serve as a band-stop filter. Similarly, when the incident light λ is at 1309nm, it would be coupled into the cavity I and II and be reflected back and forth, so little energy would be transmitted. However, while at λ=1281 nm, part of energy would be coupled into the three rectangle cavities, and SPPs can pass through the waveguide to the output port, acted as PIT “on” state.

  1. Page 5: “The height of cavity I h1 is set to 100, 110, 120, 130, and 140 nm, and the width of cavity II w2 is set to 520 to 600 nm with a 20 nm step.” can be re-wrote as “The height of cavity I h1 and the width of cavity II w2 are varied from 100 to 140 nm, 520 to 600 nm with a step of 10 nm and 20 nm, respectively.”

Reply: We have modified “The height of cavity I h1 is set to 100, 110, 120, 130, and 140 nm, and the width of cavity II w2 is set to 520 to 600 nm with a 20 nm step.” to “The height of cavity I h1 and the width of cavity II w2 are varied from 100 to 140 nm, 520 to 600 nm with a step of 10 nm and 20 nm, respectively.”

  1. Page 7: “As shown in Fig. 8, the shifting behaviors allow cavity II to partially overlap with cavity I.” Which should be Fig. 10. The cavity I, II and III should be signed in this figure as well.

Reply: We have revised it in page 7 on which cavities I, II and III have been marked.

  1. Page 7: “During the process of b increasing, the two transmission peaks also exhibit a slight red shift.” Why? What is the physical mechanism here?

Reply: The increase of parameter b is equivalent to the increase in the height of cavity II. Due to the change in the resonator structure, the wavelength of the strong coupling effect with the resonator also changes, so the two transmission peaks exhibit a slight red shift.

  1. Page 9: “The phenomenon caused by a change in the energy coupling path can be realized by changing one or two parameters in the various structures discussed in the paper.” To be revised.

Reply: We have revised “The phenomenon caused by a change in the energy coupling path can be realized by changing one or two parameters in the various structures discussed in the paper.” to “This phenomenon, caused by a change in the energy coupling path, can be realized by changing one or two parameters in the various structures discussed in the paper.”

Reviewer 2 Report

COMMENTS TO THE AUTHOR(S)

The authors have proposed numerically and theoretically the plasmonically induced transparency peaks fluctuation within multi-rectangle resonators. The induced transparency window is obtained by modifying the geometric parameters of the rectangles. However, in my opinion, the structural performance discussed by the author is not a breakthrough. I failed to get the impact and novelty of this manuscript. The impact and novelty of this paper need to be improved greatly. This paper is complicated version of the below papers without any novelties and hard for experimentally fabrication.

-       Noual A, Abouti OE, El Boudouti EH, Akjouj A, Pennec Y, Djafari-Rouhani B. Plasmonic-induced transparency in a MIM waveguide with two side-coupled cavities. Applied Physics A. 2017 Jan;123(1):1-7.

-       Yun B, Hu G, Jiawei C, Cui Y. Plasmon induced transparency in metal–insulator–metal waveguide by a stub coupled with FP resonator. Materials Research Express. 2014 Jul 1;1(3):036201.

In my opinion, this manuscript cannot be published at its present form. The following issues must be taking into account before further consideration.

1-       Based on coupled mode theory the Eq. 3 of paper is on one-resonator coupled waveguide system equation, while in your paper there are 3 coupled rectangle resonators. Please talk about this in more detail.

2-       The three main wavelengths used for fiber optic transmission are 850, 1300, and 1550 nanometers. These wavelengths are used in fiber optics because they have the lowest attenuation of the fiber. It is better that the peaks of transmission coincident with these wavelengths for communication applications.

3-       Please talk about how this device can be practically fabricated? The transmission spectra strongly depend on geometrical parameters, hence tuning of peak of transmission in particular wavelength maybe not possible because of the fabrication tolerances.

4-       Figure 6 talk about four rectangle resonator system but the forth rectangle didn’t introduce in paper.

5-       In Fig.7 for three rectangle resonator system there is five resonant dip in transmissions spectra. How this can possible physically?

6-       some latest references that are also highly related to the plasmon induced transparency were missing in introduction such as Refs: 

- Mashayekhi MZ, Abbasian K, Nurmohammadi T. Dual-wavelength active and tunable modulation at telecommunication wavelengths using graphene-metal hybrid metamaterial based on plasmon induced transparency. Physica Scripta. 2022 Aug 12;97(9):095503.

Author Response

1-Based on coupled mode theory the Eq. 3 of paper is on one-resonator coupled waveguide system equation, while in your paper there are 3 coupled rectangle resonators. Please talk about this in more detail.

Reply: Our researches in this paper are based on a single resonant cavity coupled waveguide system. Through experiments we found that if there are two resonant cavities with the same parameters above and below, the transmission spectrum does not change. But when the structure parameters of one resonant cavity are changed, the PIT window could be produced by the destructive interference between one cavity and another cavity.

2-The three main wavelengths used for fiber optic transmission are 850, 1300, and 1550 nanometers. These wavelengths are used in fiber optics because they have the lowest attenuation of the fiber. It is better that the peaks of transmission coincident with these wavelengths for communication applications.

Reply: Thanks for your suggestion. We will definitely pay attention to the three main wavelengths used for fiber optic transmission in our follow-up research work.

3-Please talk about how this device can be practically fabricated? The transmission spectra strongly depend on geometrical parameters, hence tuning of peak of transmission in particular wavelength maybe not possible because of the fabrication tolerances.

Reply: The fabrication can be completed by laser etching on the metal plate, but there is a certain difference between the two-dimensional model and the three-dimensional model, and the next step is to make a three-dimensional model of the device on this basis in order to reduce the error. In this paper, we proposed a PIT structure that achieves multiple more pronounced transparency windows. It was numerically investigated composed of a MIM bus waveguide coupled to asymmetric multirectangle resonator. A series of transmission spectrum results were obtained, which provides a certain reference for the design and research of micro/nano devices such as filters, optical switches, demultiplexers, nanosensors and so on.

4-Figure 6 talk about four rectangle resonator system but the forth rectangle didn’t introduce in paper.

Reply: Thanks for your reminder. We have put “Off-to-on transmission spectra of the four-rectangle-resonator PIT system with different values of h1 and w2.” corrected to “Off-to-on transmission spectra of the three-rectangle-resonator PIT system with different values of h1 and w2.”

5-In Fig.7 for three rectangle resonator system there is five resonant dip in transmissions spectra. How this can possible physically?

Reply: The resonator has filtering properties for different wavelengths, and the coupling effect is produced with the resonator in different bands. This phenomenon is caused by the coupling resonance between the light in the main waveguide and the different cavities.

6-some latest references that are also highly related to the plasmon induced transparency were missing in introduction such as Refs:

Mashayekhi. MZ.; Abbasian, K.; Nurmohammadi, T. Dual-wavelength active and tunable modulation at telecommunication wavelengths using graphene-metal hybrid metamaterial based on plasmon induced transparency. Physica Scripta. 2022 Aug 12;97(9):095503.

Reply: Thanks for your reminder. We have read the literature and included it as a reference for the article.

  1. Mashayekhi, M.Z.; Abbasian, K.; Nurmohammadi, T.; Dual-wavelength active and tunable modulation at telecommunication wavelengths using graphene-metal hybrid metamaterial based on plasmon induced transparency. Physica Scripta. 2022 12. 095503.

Reviewer 3 Report

The paper presents a metal-insulator-metal waveguide that exhibits plasmonically induced transparency. The transparency can be controlled by tuning the geometry of three rectangular resonators that are located near the waveguide. Numerical results are presented showing the effects of the resonator geometry on the induced transparency. The work is interesting. However, there are several issues associated with the paper that needs to be addressed before the paper can be considered for publication.

1. What finite element solver was used computer the field distributions? Was it a commercial software like Comsol multiphysics or a custom code that the authors developed? Please mention that in the manuscript with appropriate references. Also, include details about the simulation parameters.

2. It should be briefly discussed how n_eff is calculated from Eq. 1 and Eq. 2. Although this is trivial for readers with backgrounds in electrodynamics, it should nevertheless be explicitly stated in the manuscript.

3. Fig. 1 shows that there are 8 geometrical parameters that define the position and the size of the three rectangular resonators. That is a large number of parameters to tune manually. Please discuss how the values of these parameters were selected.

4. In Fig. 2, the width of the MIM waveguide is used as one of the axes. However, the width of the MIM was not defined in the text nor Fig. 1. Please correct this.

5. Have the authors considered using an optimization algorithm to tune the resonator parameters? That might be a very interesting study.

6. One of the major applications of plasmonic structures is optical trapping. The authors may choose to cite a few papers related to that topic. For example:

- https://doi.org/10.1021/nl103070n

- https://doi.org/10.1063/5.0123268

I also recommend looking into some more papers of Prof. Crozier and Prof. Hesselink.

7. The English phrasing in the paper needs to be improved (especially in the introduction and abstract sections).

Author Response

  1. What finite element solver was used computer the field distributions? Was it a commercial software like Comsol multiphysics or a custom code that the authors developed? Please mention that in the manuscript with appropriate references. Also, include details about the simulation parameters.

Reply: Comsol multiphysics software were used to realize the simulation. Add “Numerical simulations performed by the commercial finite element method (FEM) software COMSOL Multiphysics demonstrate our theoretical analysis.” in the introduction. In these simulations, it is assumed that the size of each uniform Yee cell of the structure in x and z-directions are Δx = Δz = 2 nm.

  1. It should be briefly discussed how neff is calculated from Eq. 1 and Eq. 2. Although this is trivial for readers with backgrounds in electrodynamics, it should nevertheless be explicitly stated in the manuscript.

Reply: Figure 2 depicts the real part of neff as a function of w and λ. Once w is determined, Re(neff) can be kept constant for various incident wavelengths. To elaborate on this question, we added two papers to the references.

  1. Han, Z.; Bozhevlnyi, S. Plasmon-induced transparency with detuned ultracompact Fabry-Perot resonators in integrated plasmonic devices. Opt Express, 2011, 19, 3251–3257.
  2. Tang, B.; Wang, J.; Xia, X.; Liang, X.; Song, C.; Qu, S. Plasmonic induced transparency and unidirectional control based on the waveguide structure with quadrant ring resonators. Appl. Phys. Express 2015, 8, 032202.

  1. Fig. 1 shows that there are 8 geometrical parameters that define the position and the size of the three rectangular resonators. That is a large number of parameters to tune manually. Please discuss how the values of these parameters were selected.

Reply: By reading a lot of references such as References [8, 9, 30, 44, 48] and trial and error, the values of these parameters were selected.

  1. Liu, D.D.; Sun, Y.; Mei M.F.; Wang J.C.; Pan Y.W.; Lu, J.; Fan, Q.B. Tunable Plasmonically Induced Transparency with Asymmetric Multi-rectangle Resonators. Plasmonics 2016 11, 1621–1628.

  1. In Fig. 2, the width of the MIM waveguide is used as one of the axes. However, the width of the MIM was not defined in the text nor Fig. 1. Please correct this.

Reply: We have added this setting condition “The width of the MIM waveguide is set to w=50nm” in line 79 on the second page.

  1. Have the authors considered using an optimization algorithm to tune the resonator parameters? That might be a very interesting study.

Reply: Thank you very much for your advice. We will use an optimization algorithm to tune the resonator parameters in our follow-up research work.

  1. One of the major applications of plasmonic structures is optical trapping. The authors may choose to cite a few papers related to that topic. For example:

- https://doi.org/10.1021/nl103070n

- https://doi.org/10.1063/5.0123268

Reply: Thanks for your reminder. We have read the literatures of Prof. Crozier and Prof. Hesselink and included them as a reference for the article.

  1. Yang, X.; Liu, Y.; Oulton, R.F.; Yin, X.; Zhang, X.Z. Optical forces in hybrid plasmonic waveguides. Nano Lett. 2011,11,321-328.
  2. Mohammad, A.Z.; Lambertus, H.L. Dynamically controllable plasmonic tweezers using C-shaped nano-engravings Appl. Phys. Lett. 121, 181108.

  1. The English phrasing in the paper needs to be improved (especially in the introduction and abstract sections).

Reply: We have already revised them and all the language expressions and voice were modified again.

Round 2

Reviewer 2 Report

The authors revised the manuscript according to my comments,  hence, the paper can be accepted for publications.

Reviewer 3 Report

The paper can be considered for publication. However, some of the references have not been properly formatted. For example, the last names of the authors were misidentified in ref [37]. I suggest a thorough check of the reference section by the authors and the editor office before publication.